# Bovine Ultra-Long CDR H3 Specific for Bovine Rotavirus Displays Potent Virus Neutralization and Therapeutic Effects in Infected Calves

**DOI:** 10.3390/biom15050689

**Published:** 2025-05-08

**Authors:** Qihuan Zhao, Puchen Li, Bo Wang, Baohui Li, Min Gao, Guanyi Ren, Gege Rile, Saqi Rila, Ke Ma, Fuxiang Bao

**Affiliations:** 1College of Veterinary Medicine, Inner Mongolia Agricultural University, Huhhot 010010, China; zhaoqihuan@emails.imau.edu.cn (Q.Z.); lpcc729@emails.imau.edu.cn (P.L.); wangboemail@emails.imau.edu.cn (B.W.); libaohui@emails.imau.edu.cn (B.L.); 2021202170024@emails.imau.edu.cn (M.G.); renguanyi@emails.imau.edu.cn (G.R.); 2018112172685@emails.imau.edu.cn (G.R.); 1501495097@emails.imau.edu.cn (S.R.); haruharu@emails.imau.edu.cn (K.M.); 2Key Laboratory of Clinical Diagnosis and Treatment Techniques for Animal Disease, Ministry of Agriculture and Rural Affairs, Huhhot 010010, China

**Keywords:** bovine rotavirus (BRV), ultra-long CDR H3, virus neutralization, treatment

## Abstract

Bovine rotavirus (BRV) is one of the main pathogens that cause acute diarrhea in calves under one month of age. Passive immunization has been recognized as an effective way to prevent and treat BRV infection. Recent studies have shown that 10% of bovine antibodies possess an ultra-long CDR H3 domain, which has been shown to be the smallest antigen-binding domain. Due to the extremely small size of ultra-long CDR H3 antibodies, the phage display method was utilized to obtain ultra-long CDR H3 antibodies targeting BRV, providing a new approach for the prevention and/or treatment of BRV. Here, we report the preparation of BRV-specific bovine ultra-long CDR H3 antibodies obtained by constructing and screening a phage display library containing approximately 8.55 × 10^9^ individual clones. Through three rounds of bio-panning, we identified 92 candidate clones, of which 79 exhibited specific binding activity in phage ELISAs. The recombinant bovine ultra-long CDR H3 antibodies could specifically bind to BRV in ELISAs and cell immunofluorescence assays. The neutralizing activity was further confirmed through virus neutralization tests. In the calf model experiment, the recombinant bovine ultra-long CDR H3 antibodies could relieve the symptoms of diarrhea, reduce both the amount and duration of virus release, and increase the survival in calves experimentally infected with BRV. Therefore, BRV-specific bovine ultra-long CDR H3 antibodies could serve as an effective agent for the prevention and treatment of BRV infection. At the same time, the development of ultra-long CDR H3 antibodies using phage display screening technology provides a new approach for developing biological agents for the prevention and control of infectious diseases in bovines.

## 1. Introduction

Newborn calf diarrhea (NCD) is a significant cause of morbidity and mortality in calves under 7 days of age, resulting in a reduced growth rate, increased risk of infection with other pathogens, and serious mortality and economic losses [1]. In 1969, the “reo”-like virus was first identified in the feces of calves with NCD [2]. In 1974, after observing the virus with an electron microscope, Flewett suggested naming the virus “rotavirus” as the virus particles look like wheels (given that “rota” is Latin for “wheel”). Four years later, the name was officially recognized by the International Committee on the Classification of Viruses [3]. In 1976, the virus was found in many other animals and was found to lead to acute gastroenteritis and cause serious effects in humans and animals [4].

Bovine rotavirus (BRV) is one of the main pathogens that causes acute diarrhea in calves under the age of one month [5,6]. BRV belongs to the genus rotavirus in the family *Reoviridae* and is a double-stranded ribonucleic acid virus [7]. The majority of BRVs exhibit species specificity and genetic variability arising from genetic transfer, gene rearrangements, and fragment exchanges. Its intact and infectious virions have a three-layer capsid structure [8]. The virus structural proteins include core proteins (VP1, VP2, and VP3), inner capsid proteins (VP6), and outer capsid proteins [9]. Among them, outer capsid proteins, including VP7 and VP4, can induce the production of neutralizing antibodies in animals and play an important role in the body’s immunity [10].

Passive immunization is an effective way to prevent diarrhea in newborn calves. By vaccinating pregnant cows with a BRV vaccine to induce the production of antibodies against BRV, the antibodies can be transferred to calves through lactation. To date, several BRV vaccines against the main pathogens causing calf diarrhea have been developed and used on cattle farms. The vaccines mainly contain a mixture of inactivated BRV, bovine coronavirus (BCV), and *Escherichia coli* (*E. coli*) [11]. However, the quality of bovine colostrum, especially the BRV-specific antibody titer in the colostrum, has a significant impact on the passive immunity obtained by calves. When cows are immunocompromised, the levels of key bioactive components in their colostrum may fall significantly below the established industry benchmarks or anticipated thresholds [12].

Meanwhile, an egg yolk IgY antibody against the VP8 capsid protein of bovine group A rotavirus is one of the strategies used to eliminate rotavirus infection in the animal environment and protect livestock herds. Formula milk containing an egg yolk antibody against bovine rotavirus can increase the response of mucosal antibody-secreting cells, reduce virus shedding, and reduce the severity and duration of diarrhea [13]. Vega C and others found that feeding uninfected newborn calves colostrum containing anti-BRV-specific IgY for 14 days could reduce the severity of diarrhea when the calves were challenged with BRV [14]. In 2020, a new avian IgY antibody product named IgY-DNT that can effectively prevent diarrhea in newborn calves was introduced to supplement passive immunity. IgY-DNT can regulate the intestinal mucosal immune response, resulting in an increase in antibody-secreting cells in the duodenum and ileum [15]. Nevertheless, the susceptibility of IgY to proteolysis is one of the limitations to the oral use of IgY for passive immunotherapy. The lack of standardization of the production, extraction, and purification processes for IgY antibodies from laboratory animals has been one of the major challenges in the product licensing, regulation, and approval of IgY-based bioproducts. More safety studies are needed to verify their safety as human and veterinary treatments [16].

Recent studies have revealed that approximately 10% of bovine antibodies possess a special ultra-long complementarity-determining region (CDR) H3 structure in the antibody heavy chains. This structure was first found in bovine IgM antibodies and was later also found in other bovine antibodies, such as IgG. The bovine ultra-long CDR H3 structure consists of a “stalk” structure on the surface of the antibody (a double chain with an ascending chain and descending chain) facing away from the beta ribbon, as well as a disulfide-rich “knob” structure located on the β-banded “stalk” [15]. This structure is similar to the “handle” and “cap” of mushrooms and is different from the typical antibody CDR H3 structure [17].

To understand the structure of the ultra-long CDR H3 in antibodies, Haakenson K et al. analyzed the two Fab fragments from the bovine antibodies BLV1H12 and BLV5B8 [18]. Both antibodies exhibit a unique β-stalk and knob structure stabilized by eight β-sheet hydrogen bonds (bovine-specific) [19]. The β-stalk connects to the antibody scaffold via a β-ring at its base. The descending chain contains alternating aromatic motifs (typically YXYXY), with BLV1H12 featuring “YTYNY” and BLV5B8 featuring “HSYEF” [20]. These motifs form stacked steps, enhancing β-sheet stability.

In conventional antibodies, the CDR typically serves as the contact point for binding to antigens. However, in antibodies containing a bovine ultra-long CDR H3, the CDR H1 and H2 fragments mainly play a role in supporting and stabilizing the knob structure and are not directly involved in antigen binding [21]. In most cases, only the CDR H3 in bovine ultra-long CDR H3 antibodies is used to bind antigens. Overall, the bovine immune system produces a unique CDR H3 sequence, which folds to form beta ribbon stalks and a knob structure. Changes in the number and binding mode of cysteines allow bovine antibodies to exhibit unique functions in terms of antigen recognition [22]. Previous studies have shown that antibodies with a unique bovine ultra-long CDR H3 can be used as broadly neutralizing antibodies (bnAbs) to neutralize HIV and treat HIV infection [23]. This indicates that they may also possess similar advantages against bovine pathogens such as rotavirus and foot-and-mouth disease virus. Although research on ultra-long CDR H3 antibodies has mainly focused on human diseases like HIV and cancer, their application to bovine pathogens could fill a crucial gap in the fields of veterinary medicine and agricultural biotechnology, bringing hope for the reduction in economic losses in the livestock industry.

A phage display library is a diversified phage clone population in which every clone contains a random foreign DNA insert and, hence, presents a different molecule on its surface [24]. The key advantage of phage display libraries is the possibility to test a vast number of phages in every round due to the high transformation efficiency, allowing for the identification of the most promising binders [25]. Meanwhile, phage display technology is also suitable for screening small-molecule antibody fragments, such as single-chain fragment variables (scFvs) and single-domain antibodies (sdAbs, also called nanobodies).

In our previous study, we prepared and characterized single-domain antibodies by immunizing Bactrian camels and constructing a phage display antibody library. These single-domain antibodies can be developed as a bispecific nanobody to target tumor cells or as a virus-specific binder for the detection and purification of viruses [26,27,28]. Based on our previous work, the development of bovine ultra-long CDR H3 antibodies using phage display technology appears feasible. Bovine pathogen-specific ultra-long CDR H3 antibodies may be a good candidate for the treatment and prevention of diseases since they are derived from cattle, and there is no immune reaction when applied to cattle. This could provide an effective approach for developing passive immunotherapeutics for bovine infectious diseases. This study aimed to utilize phage display antibody library technology to generate ultra-long CDR H3 antibodies that specifically target BRV, evaluate the binding and neutralizing activities of the antibodies as well as their therapeutic efficacy, and demonstrate that ultra-long CDR H3 antibodies could serve as effective agents for the prevention and treatment of bovine rotavirus infections.

## 2. Materials and Methods

### 2.1. Bovine Immunization

A five-month-old Holstein male calf was obtained from the dairy farm in the suburbs of Hohhot, Inner Mongolia Autonomous Region, and used for immunization. The calf was injected with 3 mL of the group A BRV NMG strain (concentration: 10^−6.13^/100 μL (TCID_50_); GenBank accession number MN807286.1; provided by Professor Weiguang Zhou from the College of Veterinary Medicine, Inner Mongolia Agricultural University, Hohhot, China [29]). After inactivation, it was mixed with an equal volume of sterilized Montanide ISA206 (SEPPIC, Colombes, France) adjuvant. A 5 mL volume of blood was collected from the jugular vein of the calf and allowed to stand until it naturally coagulated. Then, the serum was collected and stored at −20 °C and used as a negative control at a dilution of 1:1000. The calf was immunized 4 times in a 2-week interval. One week after each immunization, blood was collected from the jugular vein and placed at 4 °C to separate the serum for the ELISAs to evaluate the antibody titers. One week after the fourth immunization, 50 mL of blood was collected from the jugular vein of the calf, which was used to isolate peripheral blood mononuclear cells (PBMCs). The calf was housed and fed in the Laboratory Animal Center, College of Veterinary Medicine, Inner Mongolia Agricultural University, with free access to food and water. All experimental procedures were carried out in accordance with the agency and the national guidelines and regulations and were approved by the Experimental Animal Use and Care Committee of Inner Mongolia Agricultural University (approval number: NND2022023).

### 2.2. Construction and Screening of Phage Display Library

Calf PBMCs were isolated from the whole blood using bovine blood lymphocyte separation kits (Tianjin Haoyang Biological Products Technology Co., Ltd., Tianjin, China). The total RNA was extracted from the PBMCs using TRIzol reagent (Ambion, Austin, TX, USA), and RT-PCR amplification was performed using the One Step PrimeScript™ RT-PCR (Takara Bio, Shiga, Japan) reaction kit. The bovine ultra-long CDR H3 antibody fragment was amplified using nested PCR and the method used in the study by Joyce C and others, with some modifications [30]. All the primers used in this study are listed in Table 1. During the first round of PCR, primers P1 and P2, which anneal to the end of framework-3 and the head of framework-4 of the bovine immunoglobulin heavy chain, respectively, were used to amplify the fragment encompassing the complete bovine immunoglobulin heavy chain CDR H3 region. The PCR products were electrophoresed on agarose gel. The band containing the bovine ultra-long CDR H3 sequence was recovered by cutting it, extracting it from the gel, and using it as a template for the second round of PCR. The bovine ultra-long CDR H3 gene was amplified using the primer sets R311-R326. The PCR products were digested with the restriction enzymes *Nco* I and *Not* I (Takara Bio, Shiga, Japan) and ligated into the pMECS plasmid vector (a gift from Professor Serge Muyldermans, Vrije Universitaire Brussel, Belgium) with T4 ligase (Takara Bio, Shiga, Japan). The ligation product was transformed into *E. coli* TG1 competent cells (GE Healthcare, Chicago, IL, USA) to construct the antibody library. The capacity of the antibody library was estimated by calculating the number of colony-forming units on the 2×YT-AG plate containing ampicillin and glucose by diluting the antibody library in a 10-fold dilution series. The transformation efficiency and ultra-long CDR H3 gene insertion rates were assessed using PCR and the sequencing primers MP57 and GIII for the pMECS plasmid vectors.

The antibody library was added to 200 mL of 2×YT medium, and the library was cultured in a shaker at 37 °C, 250 r/min for 2 h. The M13K07 helper phage was added (NBbiolab, Chengdu, China), and the culture was incubated at 37 °C for 1 h to allow for infection and rescue. The cultured bacteria solution was centrifuged at 4000× *g* for 10 min at 4 °C, the supernatant was discarded, and the bacteria pellet was resuspended with 2×YT-AK medium containing ampicillin and kanamycin and grown at 37 °C, 220 r/min overnight. The next day, the culture medium was centrifuged at 4 °C and 7197× *g* for 15 min; 20% PEG8000 was added to the supernatant, which was then centrifuged at 7197× *g* at 4 °C for 25 min to precipitate the phages. The pellet was resuspended in PBS to obtain the recombinant phage library.

The phage library was added to 5 mL immunotubes coated with BRV at the TCID_50_ (10^−6.13^/100 μL), incubated for 1 h at 37 °C, and then the bound phages were washed with 2 mL of PBS. The bound recombinant phage was used to infect *E. coli* TG1 (OD600nm = 0.6), and the M13K07 helper phage was added for rescue. The phages were harvested, purified, and used for a new round of enrichment. The *E. coli* TG1 infected with the bound phages from the final enrichment were grown on 2×YT-AG plates. Ninety-two clones were randomly picked from the 2×YT-AG plate, added to 2×YT-AG liquid medium, and incubated at 37 °C and 250 r/min overnight. The M13K07 helper phage was added and allowed to infect for 1 h before being centrifuged at 14,000 r/min for 5 min. The bacteria pellets were resuspended in 2×YT-AK medium and incubated at 37 °C and 250 r/min overnight. The medium was centrifuged at 14,000 r/min for 5 min; 20% PEG8000 was added to the supernatant, and the mixture was centrifuged at 7197× *g* for 30 min. The recombinant phage was introduced to an ELISA plate pre-coated with BRV at the TCID_50_ (10^−6.13^/100 μL) and incubated for 2 h at room temperature to facilitate binding. The M13K07 helper phage was used as a negative control, and PBS buffer was used as a blank control. The anti-M13 bacteriophage secondary antibody (AlpSdAbs VHH, Chengdu, China) was diluted to a concentration of 1:5000 and added to each well and allowed to bind for 1 h at room temperature. A TMB solution was added for color development, and the OD405nm value was measured using a microplate reader. The absorbance of the experimental group/negative control ≥2.1 was regarded as a positive result.

### 2.3. Expression and Purification of the Bovine Ultra-Long CDR H3

The selected recombinant phages from the phage ELISA carrying the bovine ultra-long CDR H3 gene were ligated to the expression vector and transformed for expression, purification, and identification. The plasmids of the positive clones were isolated from the phage ELISA and digested with the restriction enzymes *Nco* I and *Not* I. The digested bovine ultra-long CDR H3 gene fragments were ligated into the pET-22b (+) plasmid using T4 DNA ligase. The ligation products were transformed into *E. coli* BL21 (DE3) competent cells (Sangon Biotech, Shanghai, China). The expression of recombinant ultra-long CDR H3 antibodies was induced by incubating the cells with 1 mM isopropyl β-D-1-thiogalactopyranoside (IPTG) (Solarbio Life Sciences, Beijing, China) for 12 h, and then the cells were collected and sonicated. The bacterial lysates were centrifuged, and the precipitates and supernatants were collected separately and analyzed using SDS-PAGE. Ni-NTA Sefinose™ Resin (Sangon Biotech, Shanghai, China) was used to purify the expressed recombinant ultra-long CDR H3 antibodies, and the recombinant protein was renatured with a gradient urea buffer from 8 M to 2 M and finally dialyzed in PBS after purification and analyzed using SDS-PAGE.

### 2.4. Binding Activity and Specificity of the Bovine Ultra-Long CDR H3 Antibodies

In order to determine the binding activity, the purified bovine ultra-long CDR H3 clones were diluted and added as the primary antibody to a 96-well plate coated with BRV at the TCID_50_ (10^−5.41^/100 μL). At the same time, the sonicated *E. coli* BL21 (DE3) lysate was used to transform the empty pET-22b (+) vector and used as the negative control, while PBS buffer was used as the blank control. The HRP-conjugated 6*His-tag mouse monoclonal antibody (Proteintech Group, Wuhan, China) was used as the secondary antibody at a concentration of 1:10,000. A TMB solution was added for color development, and the OD450nm value was measured using a microplate reader.

### 2.5. Immunofluorescence

MA-104 African green monkey fetal kidney cells were added to a 6-well cell culture dish at a concentration of 10^6^ cells/well and grown at 37 °C and 5% CO_2_ for 24 h. The cell culture medium was discarded, and 2 mL of DMEM medium was added to each well. The BRV was diluted to the TCID_50_ (10^−5.41^/100 μL), and 2 mL of virus diluent was added to each well, and DMEM was added to the control group. The cells were cultured in an incubator for 2 h. The medium was discarded, and 1 mL of pre-cooled ice methanol was added to each well for 20 min at room temperature to fix the cells. The fixative was aspirated, and the cells were rinsed with DPBS 3 times for 5 min each time. A 2 mL volume of a 5% bovine serum albumin (BSA) solution was added to each well, and the dish was incubated at room temperature for 1 h before 2 mL of the recombinant ultra-long CDR H3 working solution (100 μg/mL) was added to each well. Then, the dish was incubated overnight in a refrigerator. The next day, the liquid was discarded, and the cells were rinsed with Dulbecco’s Phosphate-Buffered Saline (DPBS) 3 times for 5 min each. The CoraLite^®^488-conjugated 6*His His-Tag Mouse Monoclonal antibody (Proteintech Group, Wuhan, China) was used as the secondary antibody at a concentration of 1:500; 2 mL of the working solution of the secondary antibody was added to each well, and the dish was incubated at room temperature for 1 h, and rinsed with PBST 3 times for 5 min each time. A 200 μL volume of a ready-to-use 4′, 6-diamidino-2-phenylindole (DAPI) (Beyotime Biotechnology, Shanghai, China) working solution was added to each well, and the dish was incubated at room temperature for 4 min, and rinsed with PBST 3 times for 5 min each time. The fluorescence signal was detected using a laser scanning confocal microscope (ZEISS LSM-800, Oberkochen, Germany).

### 2.6. Neutralizing Activity of the Bovine Ultra-Long CDR H3

MA-104 cells were incubated at a density of 8000 cells/well in a 96-well cell culture dish at 37 °C and 5% CO_2_ for 24 h. BRV was diluted to the TCID_50_ (10^−5.41^/100 μL), and 100 μL of virus diluent was mixed with an equal volume of purified recombinant ultra-long CDR H3 clone 84 to final concentrations of 5, 10, 20, and 40 μg/mL. The antibody and virus mixtures were incubated at 37 °C and 5% CO_2_ for 1 h and then added to the 96-well cell culture dish and incubated for 3–5 days to observe the CPEs on the cells. At the same time, PBS + BRV was used as the positive infection control, and MA-104 cells were used as the untreated, uninfected blank group.

### 2.7. The Suckling Rat Challenge Model

#### 2.7.1. BRV Challenge Test in Suckling Rats

We planned to utilize the suckling rat as a model for BRV infection to verify the effect of the recombinant ultra-long CDR H3 antibodies following similar methodologies to studies using animal models for viral infection research [31,32]. Female Wistar rats at five months of age were obtained from the Inner Mongolia Medical University. To avoid the effects of biological differences between the dams and their respective pups, the pups were shuffled before all experiments, and the litter size was adjusted to five pups per dam. The control and inoculated rats were housed separately in the Laboratory Animal Center of the College of Veterinary Medicine, Inner Mongolia Agricultural University, with free access to food and water. Five-day-old Wistar rats were weighed and divided into two groups: the BRV and PBS groups, with five rats in each group. BRV was orally administered to the BRV group at a dose of 100 μL/g (10^−5.41^/100 μL (TCID_50_)) according to the weight of the rats at 0 h. The rats in the PBS group were also orally administered with PBS at a dose of 100 μL per gram of body weight. The body weight of each rat and the occurrence of diarrhea in each group were monitored from 0 h to 156 h. The rats in the BRV and PBS groups were euthanized, and the spleen and small intestine were sampled at 24, 48, 84, 132, and 156 h.

#### 2.7.2. PCR Detection of Virus

The virus genome RNA was extracted from the spleen and small intestine of the rats using the Viral Genome DNA/RNA Extraction Kit (Spin Column Type) (TIANGEN, Beijing, China). The primers BRV-F and BRV-R (Table 1) were used to amplify the BRV nonstructural protein 5 (*NSP5*) gene [33] using the One Step PrimeScript™ RT-PCR Reaction Kit (Takara Bio, Shiga, Japan) according to the manufacturer’s instructions. The PCR products were analyzed using electrophoresis on a 2% agarose gel and purified to recover the products. These were ligated into the pMD19-T vector to construct the recombinant plasmid pMD19-T-NSP5.

#### 2.7.3. RT-qPCR

The recombinant plasmid pMD19-T-NSP5 was diluted ten-fold from 10^10^ copies to 10^1^ copies and used as a template along with the primers BRV-F and BRV-R and BRV-Probe [33] to perform RT-qPCR using the One Step PrimeScript™ RT-PCR reaction kit (Perfect Real Time) (Takara Bio, Shiga, Japan). The reaction system consisted of the following: 12.5 μL of 2×One Step RT-PCR Buffer III, 0.5 μL of ExTaq HS, 0.5 μL of PrimeScript RT Enzyme Mix II, 1 μL of BRV-Probe (10 μM), 1.0 μL each of the BRV-F/BRV-R primers (10 μM), and 2.0 μL of template, which was brought to a final volume of 25.0 μL using nuclease-free water (ddH_2_O). The RT-qPCR reaction conditions were as follows: initial reverse transcription at 42 °C for 5 min, followed by PCR activation at 95 °C for 10 s, and 45 cycles of denaturation at 95 °C for 5 s and annealing/extension at 60 °C for 30 s. After the reaction, a standard curve was plotted using Excel.

The total RNA from spleen and small intestine samples from the suckling rats was extracted and used as a template, along with the primers BRV-F and BRV-R and BRV-Probe [33], to perform RT-qPCR using the same reaction as above to obtain the Cq value and perform an absolute quantitative analysis.

#### 2.7.4. Histopathological Assessment of Rat Intestine Slices

Intestinal samples obtained from the small intestines of the rats in each group were placed in tissue embedding cassettes and immersed in a 4% paraformaldehyde (PFA) solution in a screw-cap bottle and incubated at room temperature for 24 h before rinsing with water for 8 h. The tissue samples underwent a stepwise dehydration process using graded ethanol concentrations (70%, 80%, 90%, 95%, 100%, and 100%), followed by treatment with xylene. Subsequently, the samples were infiltrated with soft wax and hard wax, each for a duration of 50 min. The tissues were cut into 5 μm thick sections with a microtome (Kedi KD-202A rotary microtome, Hangzhou, China), and the sections were floated in a water bath at 60 °C until they became smooth on the water surface before they were carefully lifted onto a glass slide (Citotest Labware Manufacturing, Jiangning, China). The glass slides were dried on a heater until the sections had fully spread out. The sections were stained with HE (Beyotime Biotechnology, Shanghai, China) and observed using an optical microscope (NIKON ECLIPSE Ts2, Tokyo, Japan) to check for intestinal epithelial cell damage, inflammation, and vacuolization in the villi.

### 2.8. Calves Challenge Model

#### 2.8.1. BRV Challenge Test and Treatment of Calves

The twelve 5-day-old Holstein male calves that were used for the BRV challenge were obtained from a dairy farm in the suburbs of Hohhot city, Inner Mongolia Autonomous Region, China, and raised in the Laboratory Animal Center of the College of Veterinary Medicine, Inner Mongolia Agricultural University. Twelve Holstein male calves were randomly divided into three groups: a BRV group (calves No. 1–4), an antibody group (calves No. 5–8.), and a PBS group (calves No. 9–12), with four calves in each group. They were fed 2 L of calf milk replacer (Inner Mongolia Knight Dairy Group Co., Ltd., Hohhot, China) twice a day.

The calves in the BRV and antibody groups were orally administered 10 mL of BRV (10^−5.41^/100 μL (TCID_50_)). The calves in the PBS group were orally administered 10 mL of PBS. After administration, the calves were observed at different time points, and the clinical symptoms were recorded. The evaluation of clinical symptoms included measuring the rectal temperature, mental state, feeding status, and occurrences of diarrhea. Fecal samples from the calves in each group were regularly collected, and the fecal score was recorded according to the scoring system shown in Appendix A. The BRV-infected calves in the antibody group received intravenous injection of the recombinant bovine ultra-long CDR H3 clones 60 and 84 at a dose of 1 mg/kg antibody protein filtered into 500 mL of physiological saline, once a day for a total of six days. The calves that died during the experimental procedure were dissected and sampled immediately after death. Except for the animals that died during the experiment, the animals in the PBS, BRV, and antibody groups in the other experiments were all euthanized at the end of the experimental period. This process was carried out in accordance with the guidelines and regulations of the relevant institutions and the state and was approved by the Experimental Animal Use and Care Committee of Inner Mongolia Agricultural University.

#### 2.8.2. RT-qPCR Detection of Virus in Calf Fecal Samples

A total of 0.2 g of the collected fecal sample was placed in 1 mL of PBS, frozen and thawed three times, filtered, and centrifuged at 12,000 r/min for 1 min to collect the supernatant. The virus genomic RNA was extracted using a virus genomic DNA/RNA extraction kit (TIANGEN, Beijing, China). The virus genomic RNA was used as a template to perform RT-qPCR (the reaction system was the same as that of the RT-qPCR system used for the Wistar rat samples) to obtain the Cq value and perform an absolute quantitative analysis. One calf from each group was randomly selected, and its RT-qPCR product was run on a 2% agarose gel.

#### 2.8.3. Stability of Anti-BRV Recombinant Bovine Ultra-Long CDR H3 Antibodies in Bovine Serum

The stability of the recombinant antibodies was assessed by incubating them with bovine serum at varying durations to monitor their binding activity to BRV using ELISAs. Briefly, 7.5 μg of the recombinant bovine ultra-long CDR H3 antibody was incubated with bovine serum at 37 °C. The incubated samples were harvested at 3, 6, 12, 24, and 48 h and frozen at −20 °C. The 0 h samples were immediately frozen at −20 °C after the addition of serum and served as the control group. The samples at each time point were used as primary antibodies and were added to 96-well Stripwell™ microplates precoated with 100 μL of BRV (10^−5.9^/100 μL (TCID_50_)). The secondary antibody was an HRP-conjugated 6*His-tagged mouse monoclonal antibody (Proteintech Group, Wuhan, China) at a concentration of 1:10,000. After adding the TMB substrate, the plate was read at OD450 nm using a microplate reader.

#### 2.8.4. Histopathological Assessment of Calf Intestine Slices

The histopathological assessment of the calves intestine slice was the same as the steps for the Wistar rat samples.

### 2.9. Statistical Analysis

GraphPad Prism 6.0 (GraphPad Software, Inc., La Jolla, CA, USA) was used for the statistical analysis. Each experiment was repeated independently three times, and the measurement data are expressed as the mean ± SD. Two-way ANOVA was used to analyze the differences between multiple groups. * *p* values < 0.05 were considered to be statistically significant.

## 3. Results

### 3.1. Immunization of Calves

The results of the ELISAs show that the OD450nm value of the basal serum was 0.1145, and upon dilution of the immune serum specific for BRV to 1:32,000, the OD450nm value remained 2.1-fold higher than the negative control, as shown in Appendix A. Therefore, we considered the titer of the immune serum to be 1:32,000.

### 3.2. Construction and Screening of Ultra-Long CDR H3 Phage Display Library

The results of the PCR show that during the initial PCR phase, the fragment containing the leader sequence to the entire CDR H3 region was successfully amplified, which revealed gene fragments that were approximately 200 bp and 150 bp in size, as shown in Figure 1A. These were used as the template for the second round of PCR to amplify the ultra-long CDR H3 fragment, which produced a fragment that was approximately 200 bp in size, as shown in Figure 1A.

The phage repertoire was calculated to have a capacity of 8.55 × 10^9^ (Figure 1B). A total of 16 clones were randomly selected from the plates for identification, and 13 clones of the target bands (400 bp) were obtained (Figure 1C). The correct insertion rate of the ultra-long CDR H3 gene fragment in the library was calculated to be 81.25%. Three rounds of bio-panning were conducted, and the BRV-specific phage clones were enriched up to 91.3 times in the third round, as shown in Appendix A.

The results of the phage ELISA are as follows. The measured OD405nm value of the negative control was 0.261, and the highest value among the selected recombinant phage clones was 1.125, with 79 clones producing OD405nm values that were 2.1-fold greater than that of the negative control (Figure 1D). The five clones with the highest absorbance values were selected and named clones 19, 46, 60, 73, and 84.

### 3.3. Expression and Purification of Ultra-Long CDR H3

The SDS-PAGE results of the recombinant antibody with an ultra-long CDR H3 after induction and purification are shown in Figure 2A. The result of the SDS-PAGE analysis of the renatured ultra-long CDR H3 antibodies showed that there was a 13 kDa protein band, which is consistent with the size of the recombinant antibody protein (Figure 2B).

### 3.4. Binding Activity and Specificity of Ultra-Long CDR H3 Antibodies

The purified recombinant ultra-long CDR H3 antibodies were used as the detection antibody in ELISAs to verify their binding to BRV. The recombinant ultra-long CDR H3 clones 19, 46, 60, 73, and 84 showed strong binding activity to BRV at concentrations of 156.25, 19.53125, 19.53125, 39.0625, and 19.53125 ng/mL, respectively (Figure 2C,D).

### 3.5. Immunofluorescence Assay

The binding of the recombinant ultra-long CDR H3 antibodies to BRV-infected cells was assessed using indirect immunofluorescence (Figure 2E). MA-104 cells infected with BRV were detected using recombinant clones 19, 60, and 84 as antibodies. However, only the results of clone 84 are presented in the images. When using 488 nm as the excitation wavelength, we found a green fluorescence signal from the cells incubated with the ultra-long CDR H3 antibody, which mainly stained the cytoplasm (Figure 2E). No fluorescence signal was observed from the control group (without the recombinant ultra-long CDR H3 antibody), as shown in Figure 2F. The results indicated that the ultra-long CDR H3 antibody may specifically bind to BRV-infected MA-104 cells.

### 3.6. Virus Neutralization Test for Ultra-Long CDR H3 Antibodies

The purified clones 19, 60, and 84 were used to treat the antibody group. MA-104 cells infected with BRV were continuously observed for 5 days under an inverted microscope, and were observed every 12 h until pathological changes occurred, after which, manual readings were carried out. The MA-104 cells in the antibody group did not show any cytopathic effects (CPEs) when incubated with the mixture of BRV and the antibodies, as shown in Figure 3A. In contrast, MA-104 cells in the control group exhibited obvious CPEs with fusiform or irregular cell morphologies, cell shrinkage, gap widening, and cell shedding, as shown in Figure 3B. Normal MA-104 cells are shown in Figure 3C. Recombinant clones 19, 60, and 84 were able to successfully neutralize ≥50% of the BRV wells at concentrations of ≥40, 20, and 20 µg/mL, as shown in Figure 3D.

### 3.7. Results of the Suckling Rat Challenge Model

#### 3.7.1. Clinical Results of BRV Challenge Test in Suckling Rats

A schematic of the BRV challenge test in suckling rats is shown in Figure 4A. After inoculating the suckling rats with BRV, the body weights of the suckling rats and the occurrence of diarrhea were measured and recorded (see Appendix A). The body weights of the suckling rats in both the BRV and PBS groups exhibited a stable increasing trend, with no significant difference observed between the two groups. The clinical observations indicated that the suckling rats in both groups were in good condition, with normal feeding and mental states, and no obvious diarrhea was observed.

#### 3.7.2. RT-qPCR Detection of BRV in Suckling Rats

A recombinant pMD19-T-NSP5 plasmid was constructed and verified to show a target band size of 127 bp (Appendix A). A standard curve was then constructed with the formula y = −3.1519x + 40.692. The correlation coefficient R^2^ was 0.995, and the amplification efficiency E was 107%. The value obtained by substituting the Cq value of the RT-qPCR results for the small intestine samples into the standard curve is shown in Figure 4B. The level of BRV in the BRV group was slightly higher than that in the PBS group, but it was not statistically significant.

#### 3.7.3. Histopathological Assessment of Rat Intestine Slices

Images of the hematoxylin–eosin (HE)-stained tissue sections of the BRV and PBS groups are shown in Figure 4. Figure 4C–E shows the duodenum, ileum, and jejunum of the small intestine of the bovine rotavirus (BRV) and PBS groups. The colors of the tissue sections from the BRV and PBS groups were similar. The intestinal epithelial cells of the duodenum, ileum, and jejunum in both the PBS and BRV groups were damaged to varying degrees, and the cell morphology had changed. The small intestinal villi in the jejunum became shorter and atrophied, and there was inflammatory cell infiltration into the ileum.

### 3.8. Results of the Calf Challenge Model

#### 3.8.1. Clinical Detection and Scoring of BRV-Challenged Calves

The protocol for administering BRV to calves to establish a calf diarrhea model is shown in Figure 5A. In this experiment, the recombinant ultra-long CDR H3 antibodies used in this experiment were a 1:1 mixture of clones 60 and 84. Compared with the other clones, they have higher binding activity and are able to neutralize BRV in in vitro experiments. The clinical observations and scoring were performed according to the methods of Buczinski S. et al. [34], and the results are shown in Figure 5B,C. The body temperature of the calves in the BRV group was higher than that of the PBS and antibody groups from day 2 to day 6 (Figure 5B). Both groups maintained relatively stable body temperatures within the normal range. The clinical score of the BRV group was higher than that of the antibody group from day 2 to day 7 (Figure 5C), indicating that the condition of the calves in the BRV group was worse than that of the calves in the antibody group, with a deteriorated mental state, elevated body temperatures, diarrhea and a reduced appetite. The calves in the BRV group died of infection on the sixth day after inoculation with BRV. These calves showed obvious symptoms of watery stools, dehydration, a poor mental state, and loss of appetite before death, as shown in Figure 5C. In the antibody group, the calves had slightly soft stools in the first two days after being infected with BRV and a slightly slower eating speed, yet their cognitive state remained unaffected. Following administration of the recombinant bovine ultra-long CDR H3 antibodies, there was a progressive amelioration of the diarrhea symptoms and a restoration of normal eating patterns. The calves in the antibody group exhibited a quicker recovery in their mental state and displayed an improved appetite following the antibody treatment. The calves in the PBS group were normal in all aspects and were in good condition (Figure 5B,C). After the challenge, the calves in the group not given the antibodies died on the sixth day, while no deaths occurred in the group given the antibodies (Appendix A).

#### 3.8.2. Results of RT-qPCR Detection of Virus in Calf Fecal Samples

Starting from the fourth day after the virus challenge, the viral loads of all the calves in the antibody group were significantly different from those in the BRV group (*p* < 0.01, *p* < 0.001) (Figure 5D). However, the BRV load in the feces of the BRV group was relatively high, and deaths occurred on the sixth day after the oral administration of BRV. BRV was not detectable in the PBS group throughout the whole experiment.

Subsequently, the fecal sample of one calf randomly selected from each group was used for PCR amplification. A 127 bp target band was detected in the samples from the antibody group from day 2 to day 4 after the BRV challenge (Appendix A). In the BRV group, a specific 127 bp PCR band was detected in all samples from day 2 to day 7, whereas no BRV was detected in the PBS group.

#### 3.8.3. Stability of Anti-BRV Recombinant Ultra-Long CDR H3 Antibodies in Bovine Serum

The results showed that the recombinant bovine ultra-long CDR H3 antibodies maintained good binding activity to BRV, and the binding activity gradually decreased over time. The binding activity was about 50% at 12 h and 25% at 24 h, decreasing to the control level at 48 h (Figure 5E).

#### 3.8.4. Histopathological Assessment

Images of HE-stained sections from the duodenum, ileum, and jejunum of the small intestine of the BRV and antibody groups are shown in Figure 6A–C, respectively. The color of the tissue sections from the BRV group was darker than that of the antibody group. In the PBS group, the intestinal villi were finger-like, and there were no inflammatory cells in the submucosa and no bleeding spots. The intestinal epithelial cells of the duodenum, ileum, and jejunum of the BRV group were damaged to varying degrees, and the cell morphology had changed. In the ileum of the BRV group, the small intestinal villi became shorter and atrophied, the size of the lymphoid nodules increased, and the number of lymphocytes increased. Some bleeding spots were also observed in the jejunum and ileum. However, the morphology of the intestinal villi was normal, and there was no obvious thickening of the submucosa and muscular layers. Compared with the BRV group, the intestinal villi in the antibody group showed some shedding, but there was no infiltration of inflammatory cells into the submucosa, and no bleeding was observed.

## 4. Discussion

Calf diarrhea is a major problem affecting the development of the cattle industry. The causes of calf diarrhea are complicated and are mainly divided into infectious and non-infectious factors. Non-infectious factors are usually attributed to the level of feeding and management. At present, large pastures provide better control over calf diarrhea caused by non-infectious factors [35]. As for infectious factors, a variety of pathogens have been identified, including rotavirus and *E. coli* [36]. Calves are highly susceptible to BRV. After being infected, the main symptom is diarrhea, accompanied by an elevated body temperature, depression, anorexia, increased salivation, abdominal pain, and diarrhea. The excreta are watery and contain mucus and blood. It can also cause various complications in the body. Moreover, calves will excrete the virus for life after infection. In severe cases, it can lead to the death of the calves [37].

Bovine ultra-long CDR H3 is the smallest antibody fragment known to be capable of binding to antigens. Compared with typical antibodies, bovine ultra-long CDR H3 antibodies have a longer complementarity-determining region, which provides a larger contact surface for antigen–antibody interactions. It can specifically recognize and bind to relatively concealed antigenic sites, whereas the Fab of traditional antibodies and scFv typically can only recognize the surface-exposed sites of the antigens. This characteristic enables bovine ultra-long CDR H3 antibodies to possess a broad and strong antigen-binding ability even in the absence of light chains. Besides the above advantages, bovine-derived ultra-long CDR H3 antibodies have a smaller structure compared to other antibodies. They can better enter and fuse with the body through the knob region. After separation, they behave like a peptide rich in disulfide bonds, and their molecular weight is approximately one-third that of a single-domain antibody. Such antibodies have a relatively strong antigen recognition ability and a relatively high affinity. Due to the high diversity and strong affinity of the bovine ultra-long CDR H3 structure [38], it has great potential in the development of new antibodies. Since the light chain of the bovine ultra-long CDR H3 does not participate in antigen binding and is only used to maintain the antibody structure, only the mispairing problem of the heavy chain needs to be addressed when constructing bispecific antibodies. Researchers developed a bispecific antibody targeting EGFR/NKp30 based on the above principle, providing a new direction for the construction of bispecific antibodies in the future [39]. Given the conformational characteristics and size (>50 aa) of the ultra-long CDR H3 region in bovine antibodies, this feature can be utilized to develop small-molecule antibodies [40].

The reason for the formation of the bovine ultra-long CDR H3 is due to the limited number of antibody gene compositions in cows compared with mice and humans, as well as other vertebrates. There are only 12 functional *VH*, 23 *DH*, and 4 *JH* gene fragments in cows compared to humans, who have 36–49 *VH*, 23 *DH*, and 6 *JH* gene fragments. In terms of diversity, theoretically, humans have as many as 6700 possible VDJ combinations, compared with just over 1000 for bovines. This makes the diversity in cows much lower, so cows develop ultra-long CDR H3 antibodies to maximize their antibody diversity [41].

The current management of BRV-induced calf diarrhea predominantly relies on antibiotic therapy. However, escalating antimicrobial resistance and regulatory restrictions on antibiotic usage have heightened treatment risks, while the absence of specific antibody-based therapeutics underscores the urgent need for targeted anti-BRV immunotherapeutics as a promising preventive strategy. In this context, bovine ultra-long CDR H3 antibodies and chicken egg yolk IgY antibodies exhibit complementary advantages in passive immunotherapy for calves. The structural configuration of bovine ultra-long CDR H3 facilitates tissue penetration, enabling deep infiltration into target tissues for therapeutic action [14]. Phage display technology allows for the efficient screening of high-affinity CDR H3 variants, supporting the rapid development of pathogen-specific therapeutic molecules against bovine pathogens. Nevertheless, the current research remains constrained to in vitro screening, preliminary in vivo models, and delivery optimization, with insufficient exploration of neutralization mechanisms in complex physiological environments and a notable absence of long-term efficacy data [30]. In contrast, chicken IgY antibodies demonstrated utility in clinical applications and scalable production. In studies by Vega C and others, oral IgY administration to a BRV infection model was found to effectively reduce the diarrhea duration, viral shedding, and clinical manifestations including fever, dehydration, and anorexia [30]. Produced through the mass immunization of hens, spray-dried IgY from egg yolk exhibits enhanced stability and significantly lower production costs compared to bovine ultra-long CDR H3 antibodies.

Bovine ultra-long CDR H3 antibodies hold substantial promise as novel therapeutics owing to their low immunogenicity and precise targeting capabilities, although their in vivo mechanisms of action warrant further elucidation. Chicken IgY antibodies, with their demonstrated clinical efficacy and cost-effectiveness, may serve as adjuncts to colostrum replacement strategies, although they will require careful consideration of the potential systemic immunosuppression risks. These two antibody platforms demonstrate complementary technical and applicational profiles, with the optimal approach contingent upon the therapeutic objectives, economic constraints, and operational environment.

In this study, the results showed that after challenge with BRV, the suckling rats did not exhibit the typical symptoms such as diarrhea or weight loss. Further analysis of the RT-qPCR data indicated that no BRV infection was detected in the small intestines and internal organs of the suckling rats. Although previous studies have successfully used suckling mice as a BRV challenge model, our results indicate that this was not the case in this study. This difference may be due to the use of different virus strains and the pathogenic properties of the virus. In view of this, we ultimately chose to directly use calves as the BRV challenge animals. The experimental results showed that the calves exhibited relatively obvious disease symptoms.

In this study, the treatment regimen involved intravenously injecting recombinant antibodies daily (1 mg/kg, once a day for 6 consecutive days). Despite the inherent instability and degradation of antibodies in bovine serum, continuous supplementation with exogenous antibodies through daily dosing ensured the maintenance of effective antibody concentrations in vivo, thus mitigating the problem of insufficient serum stability and compensating for the short half-life of the antibodies. This guaranteed that the antibody levels remained above the virus neutralization threshold during the critical period of virus replication. As can be seen from the experimental results, after the fourth day, compared with the BRV-infected calves, the antibody-treated calves showed a gradual alleviation of their diarrhea symptoms, a restoration of appetite, and a significant reduction in their viral load, as detected by RT-qPCR. These results indicate that although the serum half-life of a single antibody dose is limited, daily antibody supplementation can continuously inhibit virus replication, reduce intestinal lesions, and ultimately achieve a lasting protective effect within 6 days.

## 5. Limitations

This study provides strong evidence for exploring the therapeutic potential of anti-BRV ultra-long CDR H3 antibodies, but there are still some limitations. First, only the Inner Mongolia BRV strain was used for immunization, which may impose certain limitations on the broad-spectrum neutralization ability of these antibodies. The BRV-specific antigenic epitopes recognized by the ultra-long CDR H3 antibodies remain unclear, and further research is still needed in the future. In addition, in the large animal experiment, the small sample size may affect the reliability and reproducibility of the results.

## 6. Conclusions

This study proved that it is feasible to prepare ultra-long CDR H3 antibodies through phage display and prokaryotic expression. The preparation of BRV-specific ultra-long CDR H3 antibodies provides a new method for treating calf diarrhea caused by BRV. Ultra-long CDR H3 antibodies are bovine in origin and are thus unlikely to cause immune reactions in the body of cattle, making them ideal candidates for therapeutic agents to treat bovine infectious diseases.

## Figures and Tables

**Figure 1 biomolecules-15-00689-f001:**
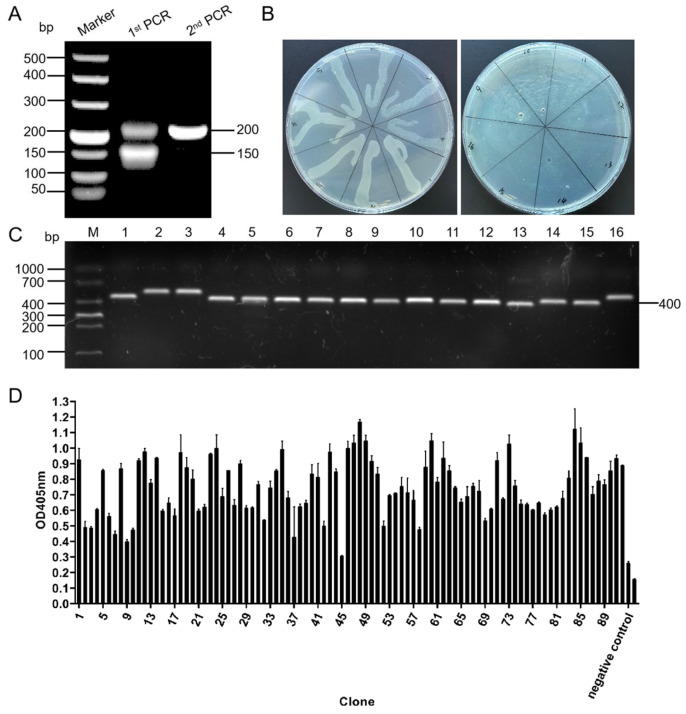
Construction and screening of phage display library. (**A**) During the construction of the phage library, an ultra-long CDR H3 gene fragment was obtained through nested PCR. (**B**) The capacity of the library was determined by growing the serially diluted constructed library on 2×YT-AG plates. (**C**) Sixteen colonies were randomly picked for PCR to determine the gene insertion rates of the library. (**D**) Ninety-two clones were randomly picked after phage-specific screening for ELISA identification.

**Figure 2 biomolecules-15-00689-f002:**
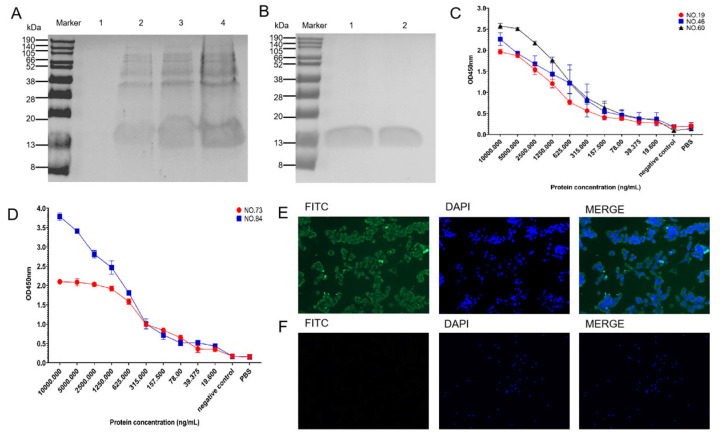
Preparation of specific ultra-long CDR H3 antibodies and detection of binding activity. (**A**) The expression of the ultra-long CDR H3 antibodies was detected using SDS-PAGE. (**B**) The purified ultra-long CDR H3 antibodies were detected using SDS-PAGE. (**C**) The binding activity of the recombinant ultra-long CDR H3 clones 19, 46, and 60 to BRV was measured using indirect ELISAs. (**D**) The binding activity of the recombinant ultra-long CDR H3 clones 73 and 84 to BRV was detected using indirect ELISAs. (**E**) The binding of the recombinant clone 84 to BRV was detected using cell immunofluorescence. (**F**) The cell immunofluorescence results of the control group (without recombinant ultra-long CDR H3 antibody).

**Figure 3 biomolecules-15-00689-f003:**
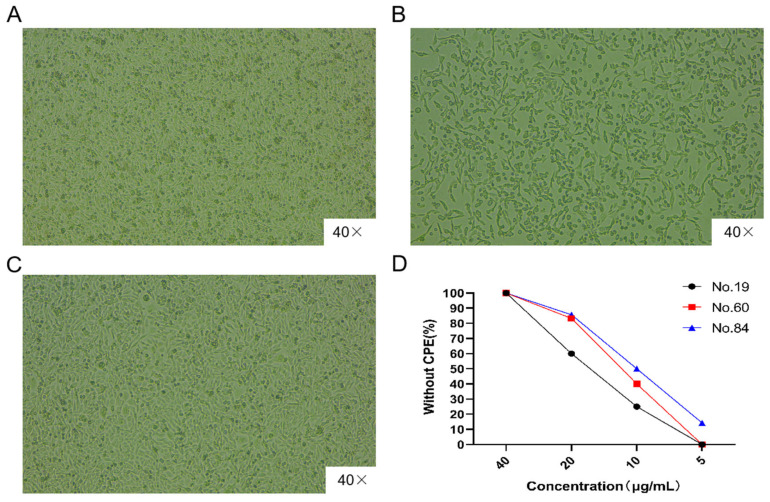
Detection of neutralizing activity of specific ultra-long CDR H3. (**A**). The results of the neutralization activity of the ultra-long CDR H3 antibody to BRV were detected using the virus neutralization test. The results showed that the cells did not develop pathological changes when the ultra-long CDR H3 antibody was present, and were the same as the normal MA-104 cells. (**B**) The results of the BRV-infected MA-104 cells in the absence of the ultra-long CDR H3 antibody. The cells showed significant cytopathic effects (CPE). (**C**) The results of normal MA-104 cells. (**D**) The recombinant ultra-long CDR H3 clones 19, 60, and 84 could neutralize the BRV in a concentration of 40 μg/mL, 20 μg/mL, and 20 μg/mL, respectively.

**Figure 4 biomolecules-15-00689-f004:**
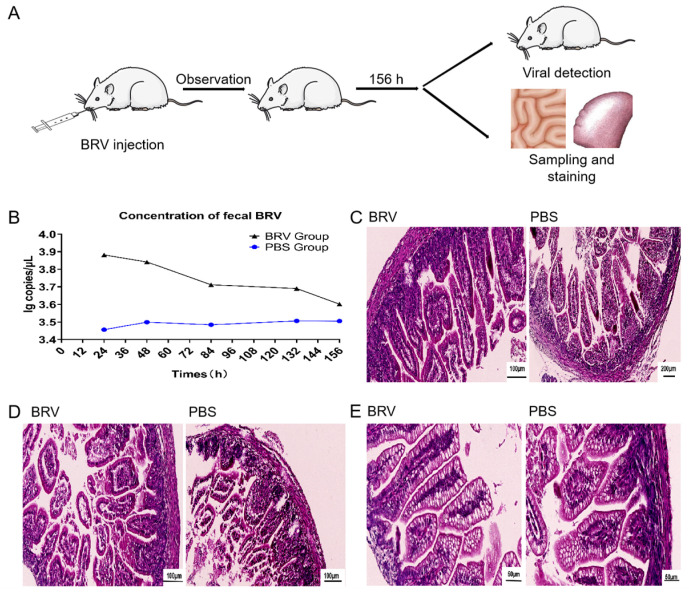
BRV challenge test in suckling rats. (**A**). The suckling rats were challenged with BRV through intragastric inoculation. (**B**) The level of BRV in the rats was detected using RT-qPCR. (**C**–**E**) The pathological changes in the BRV and control groups were observed using HE-stained paraffin sections of the duodenum (**C**), jejunum (**D**), and ileum (**E**).

**Figure 5 biomolecules-15-00689-f005:**
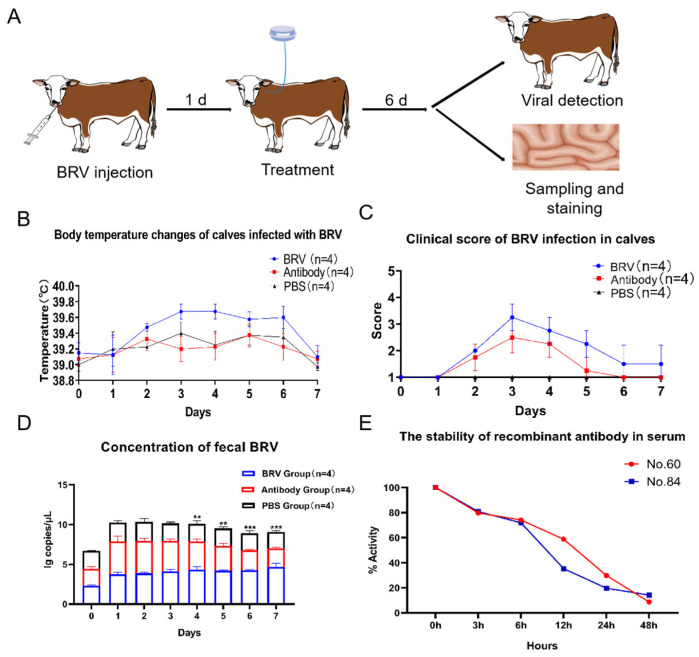
BRV challenge test in calves and the treatment effect of recombinant ultra-long CDR H3 antibodies. (**A**) After oral administration of BRV, the recombinant ultra-long CDR H3 antibodies were injected intravenously for 6 consecutive days, and their effect was evaluated. (**B**) The rectal body temperature of the calves during the experimental period. (**C**) The clinical scores of the calves during the experimental period. (**D**) The BRV level in the control group and antibody-injected calves at different time periods was detected using RT-qPCR (** *p* < 0.01, *** *p* < 0.001). (**E**) The stability of the recombinant antibody was tested by mixing the recombinant antibody with bovine serum and incubating it for different time periods, and then evaluating the changes in its binding activity to BRV using ELISAs.

**Figure 6 biomolecules-15-00689-f006:**
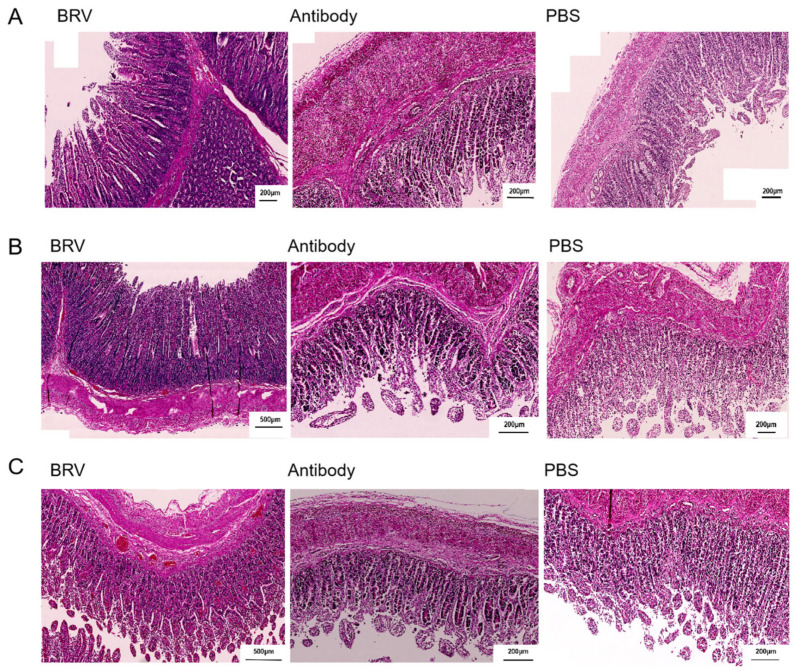
The pathological changes in the BRV, antibody, and control groups were observed using HE staining of paraffin sections of the duodenum (**A**), jejunum (**B**), and ileum (**C**).

**Table 1 biomolecules-15-00689-t001:** PCR primers.

Primer	Sequence
P1	*5′-GGACTCGGCCACMTAYTACTG-3′*
P2	*5′-GCTCGAGACGGTGAYCAG-3′*
R311	*5′-CATGCCATGGCCACTACTGTGCACCAAAAAACA-3′*
R312	*5′-CATGCCATGGCCACTACTGTGCACCAAAGAACC-3′*
R313	*5′-CATGCCATGGCCACTACTGTGCACCAAAAAACG-3′*
R314	*5′-CATGCCATGGCCACTACTGTGCACCAACAAACT-3′*
R315	*5′-CATGCCATGGCCACTACTGTGCACCAACAGACC-3′*
R316	*5′-CATGCCATGGCCACTACTGTGGTCCAGAAAACA-3′*
R317	*5′-CATGCCATGGCCACTACTGTAGTCCAACGAACA-3′*
R321	*5′-AAGGAAAAAAGCGGCCGCGGCATCGACGTACCATTCGTA-3′*
R322	*5′-AAGGAAAAAAGCGGCCGCGGTATCGACGTACCATTCGTA-3′*
R323	*5′-AAGGAAAAAAGCGGCCGCGGCTTCGACGTACAATTCGTA-3′*
R324	*5′-AAGGAAAAAAGCGGCCGCGGCATTGACGTAGAATTCGTA-3′*
R325	*5′-AAGGAAAAAAGCGGCCGCGGCCTCGATGTCAAATTCGTA-3′*
T7	*5′-TAATACGACTCACTATAGGG-3′*
T7t	*5′-TAATACGACTCACTATAGGG-3′*
MP57	*5′-TTATGCTTCCGGCTCGTATG-3′*
GIII	*5′-CCACAGACAGCCCTCATAG-3′*
BRV-probe	*ROX-CTGATTCTGCTTCAAACGATCCACTCACCAGC-BHQ2*
BRV-F	*5′-CATGYTGTCAAAGTCTCCAGA-3′*
BRV-R	*5′-TGAATCCATAGACACGCCAGC-3′*

## Data Availability

All datasets generated or analyzed during this study are available from the corresponding author on reasonable request.

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
