# Peer review of "Bovine Ultra-Long CDR H3 Specific for Bovine Rotavirus Displays Potent Virus Neutralization and Therapeutic Effects in Infected Calves"

_biomolecules, 2025, doi:10.3390/biom15050689_

Round 1
Reviewer 1 Report
Comments and Suggestions for Authors
This manuscript presents a detailed and coherent study on the development and testing of ULCDRH3 for potential use against BRV in neonatal calves. One of the key strengths of this work is the in vivo challenge model in calves, which adds real-world value and relevance and showed encouraging therapeutic potential. The use of a species-matched antibody format derived directly from bovine immune cells and the use of phage display technology to identify new ULCDRH3 brings novelty to the field, especially in the veterinary context. At the same time, there are several structural, content, and presentation issues that currently limit the readability, scientific clarity, and potential impact of the manuscript. The results section is overly segmented and includes a great deal of technical detail that should belong in the methods section, which overall makes it unnecessarily hard to follow the bigger-picture narrative and to interpret the significance of the findings. Similarly, the Discussion contains substantial redundant background information that was already partly covered in the Introduction, rather than focusing on interpretation of the results and their implications. Obviating this redundant content, the remaining discussion is notably lean and does not sufficiently engage with the study findings or the literature. Combined, these issues make the manuscript harder to follow and weaken the overall narrative.
Given the structural nature of the key issues, I have chosen to structure my review to follow the layout of the manuscript (rather than dividing my comments into "major" and "minor" points as I normally would). However, I have clearly marked the most important comments both explicitly and by color to help the authors identify them more easily.
Overall, this study likely represents a useful contribution to the field, particularly in the area of veterinary antibody therapies. However, the manuscript would greatly benefit from a major revision. I hope my feedback will be helpful to the authors in this process, and I assure them that, despite the extensive nature of my comments, my intention was to be as constructive as possible.

The manuscript is very understandable, but it could benefit from editorial polishing to improve minor syntax and grammar issues and therefore its overall readability. For linguistic issues that may affect scientific clarity, I added some feedback in the "Phrasing Issues" section of my report.
Author Response
We thank all the reviewers for their valuable comments and suggestions. We have carefully revised the manuscript to enhance its clarity and facilitate the understanding of the readers. Our point-to-point responses are presented in the following. We hope that the revision satisfactorily addresses the comments and concerns of the editors and reviewers.
Reviewer 1:
Thank you for your valuable feedback. Based on your suggestions, we have carefully addressed the identified errors in the revised manuscript and provided point-by-point responses to all your comments.
ABSTRACT:
- The methods, results and conclusions are mostlywell represented, however the aim of the study (or research question) should be more clearly stated
Response: We think that this is an excellent suggestion. We have explained the changes made, including the exact locations where these changes can be found in the revised manuscript.
- Betweenthe sentences in line 19: It would help to give some sense to how many clones were screened/obtained/tested.
Response: We have re-written this part according to the reviewer’s suggestion.
- one of thekey findings of this study (increased survival of challenged calves) is not mentioned in the abstract. This should be
Response: We have re-written this part according to the reviewer’s suggestion.
- Line25: “and proposes a new idea” – currently the grammatical subject of this sentence is the CDR H3 fragments. Therefore, this conclusion statement should be clarified and refined. Who proposes what and why?
Response: We have re-written this part according to the reviewer’s suggestion.
- It may be acceptable to summarize/removenon-critical information to compensate for slightly increased abstract length as a result of the above s
Response: Thank you for your suggestions. We have made our best efforts to polish the language in the revised manuscript.
INTRODUCTION:
- The aim ofthe study and/or research question is mostly implied rather than explicitly stated. A clearer articulation of the study objective (e.g. rephrasing lines 132-137 or adding a final paragraph or sentence along the lines of: “This study aimed to .. and evaluate...”) would help to improve the focus and emphasize the relevance/impact of the findings.
Response: Thank you for your suggestions. We have made our best efforts to polish the language in the revised manuscript.
In most cases, only CDR H3 in bovine ultra-long CDR H3 antibodies is used to bind antigens. Overall, the bovine immune system produces a unique CDR H3 sequence, which folds to form beta ribbon stalks and a knob structure. Changes in the number and binding mode of cysteines allow bovine antibodies to exhibit unique functions in terms of antigen recognition22. Previous studies have shown that antibodies with a unique bovine ultra-long CDR H3 can be used as broadly neutralizing antibodies (bnAbs) to neutralize HIV viruses and treat HIV infection23. This indicates that they may also possess similar advantages against bovine pathogens such as rotavirus and foot-and-mouth disease virus. Although research on ultra-long CDR H3 antibodies has mainly focused on human diseases like HIV and cancer, their application to bovine pathogens could fill a crucial gap in the fields of veterinary medicine and agricultural biotechnology, bringing hope for the reduction of economic losses in the livestock industry.
- Whilethe introduction is generally very detailed and complete, it could help to briefly note the relative novelty of applying ULCDR-H3 to treat bovine pathogens rather than more commonly explored uses like cancer/HIV/etc.
Response: Thank you for your suggestions. We have made changes to relevant statements in the revised manuscript.
- Lines86-109: this section provides a well-crafted and detailed overview of UL CDR H3 structural features and However, given this study did not address any structural or mechanistical features, this long technical section seems unnecessary. I encourage the authors to greatly summarize it, and/or refer to a review article, and/or include a schematic as visual that replaces part of the text. Note that reducing this section would be important, given my recommendations about moving large part of the discussion to the introduction (see further below).
Response: We have re-written this part according to the reviewer’s suggestion.
RESULTS:
Major Comment:
- A major concern with the Results section is its structural organization and narrative style. There is an excessive number of subsections, which also contain extended descriptions of technical procedures, such as primer names, gel electrophoresis conditions, vector construction, etc, which should belong in the methods section. This not only deviates from convention but also greatly dilutes the readability and the impact of the results section, making it read more like a methods section. I strongly recommend a major revision of the entire Results section to focus on reporting and interpreting the conclusion-relevant findings, while moving most unnecessary procedural details to the methods section (if already covered there, then no need to repeat it). Coalescing unnecessary sub-sections, improving data presentation, and summarizing key findings after objective narration of the results, would significantly improve the quality of the manuscript. Just to give one example: all subsections between lines 242-280 could be consolidated into a single section (suckling rat challenge model). Unnecessary detail should be moved to the methods section and leave only a concise but sufficient description of the investigational approach and the experimental findings. This section in particular should be brief, given the findings are not conclusive since the infection model was not validated.
Response: We have re-written this part according to the reviewer’s suggestion.
- In many results paragraphs, the first sentence(s) introduces a figure and provides excessive detail that should belong in the figure Conversely, a helpful “ (Fig.X)” reference is missing at the end of many key results-description statements. As they restructure the results section, I encourage the authors to improve this aspect as well, and focus on a coherent results-driven narrative rather than sub-sectioning based on methods or figures. Introducing figures in the main text is not necessary, since this is what the figure legends are for. However, it would help to add (Fig.X) at the end of relevant statements.
Response: We have re-written this part according to the reviewer’s suggestion.
- Line 182-192: Ultra-long CDR H3 antibody fragments are described in extensive detail in the introduction and the However, when the authors describe their experimental approach (in the results and methods sections), it is not immediately clear to the reader what type of protein is expressed as a consequence of ligating the ULCDR-H3 fragment into the pET-22b vector (e.g. whole antibody vs. single-domain structures, or something else). This can besomewhat inferred from the 13-kDa bands mentioned in lines 191-192) but clarity to the reader is still lacking, especially given reference to “consistent with the expected result as shown in Figure 2B” (it is not clear what the expected results are, nor what they are based on, nor how this is related to the picture of agar plates in figure 2B). Also, regarding “subsequent antibody engineering” (line 19). If the expressed protein products includes any additional scaffold, tags, fusion-proteins, or other supporting domains this should also be specified somewhere in the results narrative section.
Response: We have re-written this part according to the reviewer’s suggestion.
Figure 2B does not mention the plate. The plate image might be Figure 1B. The expressed protein product is a recombinant protein formed by the ligation of the ultra-long CDR H3 and pET-22b(+). Since it includes the HIS tag carried by the pET-22b(+) vector, the tag is not listed.
- Line 201-211 and 409-410: while it is a logical interpretation of the results, a definitive conclusion that the tested ULCDRH3 specifically binds to BRV is not entirely supported by the data. Given the use of a secondary antibody staining method, it is unproven whether the IF signal is coming from binding of the primary antibody to a viral protein. Therefore, these conclusive statements should be softened as “ indicates that ULCDRH3 likely binds to …” or similar
Response: We have re-written this part according to the reviewer’s suggestion.
- Figure 2: please double-check this figure, since letters in the panel (e.g. B, C etc) do not match what is shown in the figure. Also, it should be clarified whether all clones were tested for immunofluorescence and which clone do the shown images correspond to?
Response: We have re-written this part according to the reviewer’s suggestion.
MA-104 cells infected with Bovine Rotavirus (BRV) were detected using recombinant clones 19, 60 and 84 as antibodies. However, only the results of clone 84 were presented in the images.
- Figure4 (lines 275-279): This could be consolidated as: The pathological changes in the BRV group and control group of the rats were observed by using H&E-stained paraffin sections of the duodenum (C), jejunum (D) and ileum (E).
Response: Thank you for your suggestions. We have made our best efforts to polish the language in the revised manuscript.
- Neutralizationresults (Line 221-241):
- this is perhaps the only section that suffers from the inverse problem (not enough methodological detail provided in order to allow the reader to correctly interpret the results). The neutralization assay should first be briefly described to orientate the reader (see comments about this assay in the methods section of this report)
Response: Thank you for your kind suggestions. We have included some additional text in the revised manuscript to describe the neutralization assay.
- Line 224: “the cells were continuously observed for 5 days” – but when was the readout/scoring officially performed? This should be specified in the text and/or Figure3 and/or Table
Response: The cells were continuously observed for five days, and observed every 12 hours. Manual readings were carried out after pathological changes occurred.
- Line 229: it is not clear why only 3 of the 5 clones were tested in the neutralization assay. Why these ones?
Response: During the experiment, a preliminary experiment was first conducted. It was found that, among the five clones, while the ELISA binding activities of the other two clones were very high, their neutralizing activities were extremely low (almost negligible). Therefore, there were no data to support them.
- Just as a comment: It is unfortunate that the authors did not test sufficient dilutions to get a full neutralization curve that would allow estimating the IC50 value. This is normally critical for this type of However, given the convincing in vivo experiments, it is not a critical flaw in this manuscript but it would have been a very valuable add-on.
- Table 1: this table is acceptable but not reader-friendly. A graph with “curves” would be much more helpful (despite the few dilutions tested), or alternatively a simple grid-like plate representation that shows the CPE-positive/negative wells in different colors (this would be more visual and easier to rapidly interpret).
Response: Thanks for the suggestions. We have included a figure of the percentage of CPE in the revised manuscript.
- Line 240: what is meant by “could neutralize”? I would suggest phrasing this as: could neutralize ≥50% of wells at concentrations ≥X µg/mL
Response: Recombinant clones 19, 60, and 84 were able to successfully neutralize ≥50% of the BRV wells at concentrations of ≥40, 20, 20 µg/mL, as shown in Table 1.
- Line269-270: how do the authors explain/interpre that the uninfected control group shows intestinal damage?
Response: During the experiment, a preliminary experiment was first conducted. It was found that, among the five clones, while the ELISA binding activities of the other two clones were very high, their neutralizing activities were extremely low (almost negligible). Therefore, there were no data to support them.
- Bovinechallengemodel (Lines 280-361):
- This section should specify which ULCDRH3 Clone was used for treatmentand why.
Response: In this part of the experiment, the recombinant ultra-long CDR H3 antibodies used were clones 60 and 84. Compared with other clones, they were found to have higher binding activity and were able to neutralize BRV in the in vitro experiments. Therefore, they were selected for the subsequent model tests.
- Line281-282: the three treatment groupsshould be introduced with clear definitions, and it should be specified how many calves were in each group. Note that currently there is ambiguity in this section regarding the groups. This is because the BRV group is not the only group that was exposed to BRV, and because the PBS group could have received PBS in lieu of the virus or in lieu of the antibody.
Response: The number of groups and other details in this part have been specified in the Methods section.
iii. Line 281-298: please add (Fig.X) at the end of each relevant statement.
Response: Thank you for your suggestions. We have made our best efforts to polish the language in the revised manuscript.
- Line 286: a reference ideally should be provided for this clinical scoring method. If this is not possible, the authors should first briefly introduce the rationale behind the scoring method before describing the results.
Response: We have re-written this part according to the reviewer’s suggestion, and supplemented it in Table 1 of the supplementary materials.
- Line286-2391: whyis the clinical score of the PBS group not described? According to Figure 5C, this group had a clinical score of zero throughout the experiment. How do authors explain/interpret this?
Response: As shown in Figure 5C, the clinical score of the PBS group is not 0 but 1, which indicates that the experimental operation itself did not induce a persistent pathological state. The transient stress response in the PBS group (e.g., reduced activity in the few hours after injection) was classified as 'mild', due to its reversibility.
- Line289-291:it would be critical to specify how many calves died in each group and also this very important data is missing in Figure 5.A good way to present this data would be through a Kapplan-Meier plot.
Response: Thank you for your suggestions. We have made our best efforts to polish the language in the revised manuscript.
- Line 203: What is meant by “all the antibody groups”?
Response: Starting from the fourth day after the virus challenge, compared with the BRV group, the viral loads of all calves in the antibody group were significantly different from those in the BRV group, and the differences were statistically significant (p < 0.01, p < 0.001).
- Line 306: “As for the PBS group, BRV was not detectable in the PBS group throughout the whole experiment by RT-qPCR detection.” – Is this referring to Figure 5D? The PBS group seems to have the highes viral load here.
Response: Based on the presented data, fecal BRV concentrations in the PBS group remained relatively stable over time. Although these values exceeded the lower limit of detection (approximately 3.4 log₁₀ copies/μL), this observed positivity in the PBS group may represent false signals which are attributable to the high sensitivity of RT-qPCR or potential laboratory contamination. In contrast, the BRV group exhibited slightly higher initial fecal BRV concentrations, when compared to the PBS group. However, when benchmarked against subsequent calf challenge data (not shown here), these values likely still fall within the negative range, as no significant temporal changes were observed. Throughout the observation period (12–156 h), BRV concentrations in both groups remained comparable, demonstrating stable and plateaued trajectories.
- Line 324-326: given the rapid destabilization of the antibody in bovine serum, how do the authors explain/interpret continued in vivo protection up to 6 days post-infection? This should be at least mentioned in the
Response: In this study, the treatment regimen involved intravenously injecting recombinant antibodies daily (1 mg/kg, once a day for 6 consecutive days). Despite the inherent instability and degradation of antibodies in bovine serum, continuous supplementation with exogenous antibodies through daily dosing ensured the maintenance of effective antibody concentrations in vivo, thus alleviating the problem of insufficient serum stability and compensating for the short half-life of antibodies. This guaranteed that the antibody levels remained above the virus neutralization threshold during the critical period of virus replication. As can be seen from the experimental results, after the fourth day, compared with the BRV-infected calves, the antibody-treated calves showed a gradual alleviation of diarrhea symptoms, a restoration of appetite, and a significant reduction in viral load, as detected by reverse transcription quantitative polymerase chain reaction (RT-qPCR). These results indicate that although the serum half-life of a single antibody dose is limited, daily antibody supplementation can continuously inhibit viral replication, reduce intestinal lesions, and ultimately achieve a lasting protective effect within 6 days.
- In Figure 5, either the legend or the graph should show how many calves were in each group.
Response: Thank you for your suggestions. We have made our best efforts to polish the language in the revised manuscript.
- In Figure 5, if calves were injected with antibody every day for 6 days (as suggested in line 605 in the methods section) this should be reflected in the schematic and in the figure legend.
Response: Thank you for your suggestions. We have made our best efforts to polish the language in the revised manuscript.
- Line342-352:please include (Fig.X) at the end of each relevant statement to guide readers to the corresponding panels of Figure 6.
Response: We have re-written this part according to the reviewer’s suggestion.
- Figure 6:the entirefigure legend is incorrect. It seems to describe elements of Figure 5.
Response: We have re-written this part according to the reviewer’s suggestion.
DISCUSSION:
- Major Comment:Whilethe information provided in Linea 362-402 is well-written and informative, this content largely overlaps with content presented in the Introduction and more importantly it is not directly related to or indirectly relevant to the results generated in this study. Therefore, I strongly encourage the authors to remove most (if not all) of this content from the discussion and consolidate it with the relevant sections in the introduction. With the removal of the redundant background section, the remainder of the Discussion is disproportionately short and does not provide a sufficient depth of analysis. The authors should substantially expand this section considering (but not limited to) the suggestions below:
- The discussion section should:
- brieflyrecapitulatekey findings and prioritize their interpretation in light of the existing relevant literature.
- Highlight strengths and limitations of the study
- Modestly emphasize the novelty aspect of the methods/findings/analysis that merit publication of this study as a sufficiently original and impactful contribution to its field.
- Discuss or explore the implications for future studies and/or the current state of the field
Response: We have re-written this part according to the reviewer’s suggestion.
- Limitations:Currently, no limitations of the study/methodology are discussed. Some that should be considered include:
- A specific strain of BRV was used for immunization, however it is unclear which viral strain was used for the neutralization assay. To demonstrate true therapeutic potential, the antibody should ideally have been tested against a panel of different strains based on their global prevalence or local distribution.
- No mechanisitic exploration or epitome mapping was attempted. Itiscurrently unclear which viral protein is bound by the antibody. This would be very helpful to know, as it could enable use of the antibody for diagnostic/research purposes, and also because on could bioinformatically assess the level of conservation of this protein across different strains (to predict cross-neutralization of different strains, given the lack of in vitro data).
iii. Limited statistical power for in vivo challenge model: I must first note that it is perfectly understandable why larger calve groups are logistically challenging, and I applaud the authors’ efforts and success in obtaining the dataset presented, which is very convincing. However, a group size of four calves does limit statistical power, and this limitation should be at liest briefly acknowledged.
- Theneutralization steps performed in this study were insufficient to obtain full neutralization curves that would allow to calculate IC50 values. Since these values are most commonly used for comparing the potency and utility of different neutralizing agents, this is a small but important limitation of this study.
Response: We have re-written this part according to the reviewer’s suggestion.
- Otherpotentially discussion-worthy topics (especially when relevant literature is available):
- Reflections on potential causesforthe failure of the suckling rat model.
- Comparison ofULCDRH3 features(e.g. neutralization potency, serum stability, cost of manufacturing) with those of known standards or similar therapeutic candidates (e.g. IgY).
- SuitabilityofULCDRH3 for practical veterinary use, e.g., route of administration, dosing regimen, cost-effectiveness, likelihood of resistance development, veterinary regulatory landscape, etc.
Response: We have re-written this part according to the reviewer’s suggestion.
METHODS:
- Line434-440: Was the virus material used for immunization previously inactivated? If so, this should be specified.
Response: We sincerely appreciate your valuable suggestions. The virus mentioned in the text regarding the issue you raised is an inactivated virus.
- Line484 and 496: which strain was used for biopanning and ELISA?
Response: The strain mentioned in the article is an ultra-long CDR H3 antibody library, obtained through the screening process with phage packaging.
- Table 2: PCRprimers – this would sit better in the supplementary (optional suggestion)
Response: Thank you for your suggestions.
- Lines552-560 (Neutralization assay):
- Which strain was used for neutralization testing and why?
Response: BRV was diluted to a TCID50 of 10-5.41/100 μL, using a group A BRV NMG strain (GenBank accession number MN807286.1)
- Line 556:were this the concentrations of the antibody prior to mixing with the virus or the final antibody concentrations in the neutralization experiment?
Response: Thank you for pointing this out. The antibody concentration in the article is the gradient concentration set before mixing with the virus.
- Which day after infectionwere the plates scored for CPE?
Response: The answers to the questions raised by the reviewers are as follows. The observation was carried out 3 to 5 days after infection, and CPE (Cytopathic Effect) scoring was conducted after the occurrence of lesions.
- Line559: suggest repffhrasing as: “PBS+BRV was set as the positive infection control and MA-104 cells were set as the untreated, uninfected blank group.
Response: We have re-written this part according to the reviewer’s suggestion.
- Line609: which dierent timepoints, and who did the clinical scoring?
Response: The calf status scoring form mentioned in the article was tested and recorded every morning during the detection cycle, and the testing and scoring were jointly carried out by the authors (Qihuan Zhao, Puchen Li, Bowang, etc).
- Isthere a reference/precedent for this clinical scoring system? Or was this developed by the authors?
Response: We referred to the scoring criteria in other articles, which has been further elaborated in the article.
- Line 657:“each experiment was independently repeated 3 times” – please revisit this statement and clarify which experiments this applies to (or which are the exceptions).
Response: During the construction of the phage antibody library, the enrichment of the antibody library was followed by repeated screening using phage ELISA after each round of enrichment. Therefore, repeated experiments were carried out.
CONCLUSIONs:
- Line662-667: after the major revision of the results and discussion sections has been finalized, this conclusions section should be revisited to ensure that the conclusions are:
- Directly related to and supported by the findings in this study, and/or
- Been sufficiently discussed in the discussion incl. comparison to relevant literature, and/or
- Not overstated and clearly distinguishing between implications for current practice for future work
Response: We have re-written this part according to the reviewer’s suggestion.
PHRASING ISSUES:
- Line58: “Sometimes the quality of bovine colostrum is not as good as expected because of the cow's own poor immune status” – Both the scientific language and clarity of this statement could be improved.
Response: We have re-written this part according to the reviewer’s suggestion.
- Line125: “the most promising binders” - should be refined to improve scientific clarity.
Response: We have re-written this part according to the reviewer’s suggestion.
- Line125-127: “also more conductive to” is slightly unclear. Do the authors mean well-suited to? Also, "small-molecule antibody fragments" is confusing, given that the listed examples are not “small molecules” in the pharmacological sense.
Response: We agree that these phrases should be modified, and appreciate your help in pointing this out.
- Line117-120: the authors mention “previous studies”, however only 1 article is cited. It would be more helpful to make this statement braoder (briefly covering the key main intended applications of ULCDRH3 in other areas like cancer, HIV, etc) and cite a good review.
Response: We are sorry for our careless mistake in this regard. Thank you for your reminder.
- Line198: “showed strong binding activity ” – it is not clear “strong ” compared to what, since no control antibodies/reagents were included in this One way to support this statement would be to comment on whether the presented ng/mL concentrations are in the nanomolar/picomolar range, considering ULCDR3H molecular weight.
Response: Thank you for your careful review. We are truly sorry, and have made corrections in accordance with your comments.
- Line218: “The results of the binding of the recombinant ultra-long CDR H3 tBRV were detected by cell immunofluorescence”. This is confusing and Perhaps fuck better: “Binding of the recombinant ultra-long CDR H3 to BRV-infected cells was assessed by indirect immunofluorescence”
Response: Again, we are sorry for our careless mistakes. Thank you for your reminder.
- Line284: suggest considering rephrasing as: “… as shown in Figure 5B, but both groups maintained …”
Response: We appreciate your help in pointing this out, and have modified the phrasing accordingly.
- Line288:“are worse than calves in the Ab group in terms of…” – this could be improved for scientific language and clarity. E.g.: “showed deteriorated mental state, increased body temperatures, diarrhea, and decreased appetite”.
Response: We agree that this phrasing should be modified, and appreciate your help in pointing this out.
Reviewer 2 Report
Comments and Suggestions for Authors
The article mainly expressed bovine ultra-long CDR H3 specific for bovine rotavirus displays potent virus neutralization and therapeutic effect. The scheme provided a novel way to cure bovine rotavirus. At present lactogenic immunity is an effective way to prevent diarrhea in newborn calves by vaccinating pregnant cows with BRV vaccine. It existed poor quality. The new idea in the article supplemented with the effect of traditional vaccine. The conclusions was consistent with the evidence and arguments presented. In this research, I agree to accept it after minor revision. There are four points needed to be revised: First the article need to be rewrited by a native speaker. Second scale bar was missing in figure 3. Third the caption of figure 6 was wrong. Fourth all the TCID50 in the article was expressed by mistake.
Author Response
We thank all the reviewers for their valuable comments and suggestions. We have carefully revised the manuscript to enhance its clarity and facilitate the understanding of the readers. Our point-to-point responses are presented in the following. We hope that the revisions made satisfactorily address the comments and concerns of the editors and reviewers.
Reviewer 2:
Thank you for your valuable feedback. Based on your suggestions, we have carefully addressed the identified errors in the revised manuscript and provided point-by-point responses to all of your comments.
First the article need to be rewrited by a native speaker.
Thanks for your suggestion. We have tried our best to polish the language in the revised manuscript
Second scale bar was missing in figure 3.
Response: We sincerely appreciate your professional evaluation of the article, and the corresponding revisions have been made in the main text. We are extremely grateful for your assistance.
Third the caption of figure 6 was wrong.
Response: We are sorry for our carelessness. In our resubmitted manuscript, this text has been revised. Thank you for your correction.
Figure 6. The pathological changes in the BRV, antibody and control groups were observed using HE staining of paraffin sections of the duodenum (A), jejunum (B) and ileum (C).
Fourth all the TCID50 in the article was expressed by mistake.
Response: We sincerely thank the reviewer for their careful reading. As suggested by the reviewer, we have corrected “TCID50””to “TCID50”.
Reviewer 3 Report
Comments and Suggestions for Authors
The manuscript entitled Bovine ultra-long CDR H3 specific for bovine rotavirus displays potent virus neutralization and therapeutic effect in infected calves requires a lot of work on style and form. Some of its methods are not clear, nor does it specify the use of animals in experimentation under an animal welfare vision. It could have declared the animals as experimental death and not let them die. It seems serious to me that the rules and regulations for the use of animals in experimentation were not applied.
LINE 33-34: Reference source not found
LINE 38: Reference source not found
LINE 56: scientific names in italics
LINE 60: ¿yolk?
LINE 65: Correct citation form, “Vega C et al.” It is not correct.
LINES 161-162: how did the authors calculate the insertion rate?
In Figure 1A two bands are shown, please indicate what are the molecular weights of these bands, since in the text it indicates one band and in the figure two are shown.
The figure caption (2B) does not correspond to the figure. Figure 2B is from an SDS-PAGE.
Figure captions (2C) and (2D) do not correspond to the figures. Figures 2C and 2D are indirect ELISA plots.
Figure caption (2E) does not correspond to the figure.
FIGURE 2F has no figure caption.
LINES 228-230 and 240-241: why do they mention that clones #60 and #84 successfully neutralized BRV at a concentration of 20ug/mL when Table 1 shows a percentage inhibition of the cytopathic effect of 83.33% and 85.71% for clones 60 and 84, respectively. Additionally, in all clones the inhibition of the cytopathic effect was 100% at a concentration of 40 μg/mL?
LINES 261-262: Did the PBS (control) group also show amplification for BRV? So what difference is there between the two groups, if the control also tested positive?
LINES 268-270: If damage was also observed in the control (PBS), how do you know that it was really BVR that caused the lesions in the organs? So what is the difference between both groups, if the control also tested positive? What do you associate that both groups showed damage in the intestine?
LINES 289-290: were they not given any treatment for dehydration or pain management? Did all the animals in the group die? Were the animals left to die without any animal welfare? This is important to mention, since in your methodology you do not mention it. Although an ethics committee supposedly approved the protocol, there is NO SCIENTIFIC JUSTIFICATION for letting the animals die without humane slaughter.
LINE 318 “in vitro” italics
LINES 354-361: The figure caption does not correspond with the images, when items A, B and C show different organ sections (duodenum, ileum, and jejunum) of the three experimental groups (BRV, PBS and Antibody)
LINE 368 “E.coli” italics
LINES 404-405: Logarithmic numbers are expressed in superscript, please correct this.
LINES 408: Scientific names are in italics.
LINES 408-418: not discussion. Please restructure the discussion
LINES 419-421 This is also mentioned in the methodology, mentioning it up to this point causes confusion.
LINES 433-438 was it inactivated, attenuated or live virus?
LINE 457 Correct the way of citing references
LINE 486 The scientific names are in italics.
LINE 488 Scientific names are in italics
Subsection 4.4: Where are the results of this section?
Subsection 4.7: Lines 419-421 could be placed here.
LINE 568: how was the PBS administered? It is important to mention how it was administered and the amount administered.
Subsection 4.8: the results of this section?
LINE 578 Latin terms are in italics
Subsection 4.10. There is no mention of this test being approved under an ethics committee, is it the same as lines 445-450?
LINES 628-629 Were all animals in the experimental groups humanely killed or how many were killed? It is important to mention this and whether it was approved under an ethics committee.
LINES 651-652 Is this antibody anti-bovine? It is also important to mention the time periods in which the stability of the recombinant antibody was monitored.
LINES 658-660 It is important to specify which variables or tests were statistically analyzed.
Conclusions: The authors could calculate the reduction in the incidence of diarrhea or symptoms in order to give more relevance as a therapeutic to their recombinant protein.
Author Response
We thank all the reviewers for their valuable comments and suggestions. We have carefully revised the manuscript to enhance its clarity and facilitate the understanding of the readers. Our point-to-point responses are presented in the following. We hope that the revisions made satisfactorily address the comments and concerns of the editors and reviewers.
Reviewer 3:
We sincerely thank the reviewers for their valuable feedback. We have improved the quality of our manuscript based on these comments. The reviewers' comments are listed below in bold font, while our responses are presented in regular font, and the modifications or additions made to the manuscript are shown in red bold text.
1.LINE 33-34: Reference source not found
2.LINE 38: Reference source not found
Response: We are sorry for our carelessness. In our resubmitted manuscript, these errors have been addressed. Thank you for your correction.
Newborn calf diarrhea (NCD) is a significant cause of morbidity and mortality in calves under 7 days of age, resulting in a reduced growth rate, increased risk of infection with other pathogens, and serious mortality and economic losses1. In 1969, the "reo" -like virus was first identified in the feces of calves with NCD2. In 1974, after observing the virus with an electron microscope, Flewett suggested naming the virus "rotavirus" as the virus particles look like wheels (given that "rota" is Latin for "wheel"). Four years later, the name was officially recognized by the International Committee on the Classification of Viruses3. In 1976, the virus was found in many other animals, and was found to lead to acute gastroenteritis and cause serious effects in human and animals4.
3.LINE 56: scientific names in italics
Response: We are sorry for our careless mistakes; thank you for your reminder.
Passive immunization is an effective way to prevent diarrhea in newborn calves. By vaccinating pregnant cows with a BRV vaccine to induce the production of antibodies against BRV, the antibodies can be transferred to calves through lactation. To date, several BRV vaccines against the main pathogens causing calf diarrhea have been developed and used on cattle farms. The vaccines mainly contain a mixture of inactivated BRV, bovine coronavirus (BCV) and Escherichia coli (E. coli)11.
4.LINE 60: ¿yolk?
Response: We are sorry for our carelessness. In our resubmitted manuscript, this text has been revised. Thank you for your correction.
Meanwhile, an egg yolk IgY antibody against the VP8 capsid protein of bovine group A rotavirus is one of the strategies used to eliminate rotavirus infection in the animal environment and protect livestock herds.
5.LINE 65: Correct citation form, “Vega C et al.” It is not correct.
Response: We sincerely thank the reviewer for their careful reading of the text. As suggested by the reviewer, we have corrected“Vega C et al.” to “Vega C and others”, as follows:
Vega C and others found that feeding uninfected newborn calves colostrum containing anti-BRV-specific IgY for 14 days could reduce the severity of diarrhea when the calves were challenged with BRV14.
6.LINES 161-162: how did the authors calculate the insertion rate?
Response: The correct insertion rate of the ultra-long CDR H3 gene fragments in the library was determined to be 81.25% by comparing the number of clones verified by electrophoresis (13 clones) with the total number of clones (16 clones).
7.In Figure 1A two bands are shown, please indicate what are the molecular weights of these bands, since in the text it indicates one band and in the figure two are shown
Response: We think that this is an excellent suggestion. We have explained the changes made, including the exact location where they can be found in the revised manuscript.
During the initial PCR phase, the fragment containing the leader sequence to the entire CDR H3 region was successfully amplified, which revealed gene fragments that were approximately 200 bp and 150 bp in size, as shown in Figure 1A. These were used as the template for the second round of PCR to amplify the ultra-long CDR H3 fragment, which produced a fragment that was approximately 200 bp in size, as shown in Figure 1A.
8.The figure caption (2B) does not correspond to the figure. Figure 2B is from an SDS-PAGE.
9.Figure captions (2C) and (2D) do not correspond to the figures. Figures 2C and 2D are indirect ELISA plots.
10.Figure caption (2E) does not correspond to the figure.
11.FIGURE 2F has no figure caption.
Response: The caption for Figure 2 has been modified as follows:
Figure 2. Preparation of specific ultra-long CDR H3 antibodies and detection of binding activity. (A) The expression of the ultra-long CDR H3 antibodies was detected using SDS-PAGE. (B) The purified ultra-long CDR H3 antibodies were detected using SDS-PAGE. (C) The binding activity of the recombinant ultra-long CDR H3 clones 19, 46 and 60 to BRV was measure using indirect ELISAs. (D) The binding activity of the recombinant ultra-long CDR H3 clones 73 and 84 to BRV was detected using indirect ELISAs. (E) The binding of the recombinant clone 84 to BRV was detected using cell immunofluorescence. (F) The cell immunofluorescence results of the control group (without recombinant ultra-long CDR H3 antibody).
12.LINES 228-230 and 240-241: why do they mention that clones #60 and #84 successfully neutralized BRV at a concentration of 20ug/mL when Table 1 shows a percentage inhibition of the cytopathic effect of 83.33% and 85.71% for clones 60 and 84, respectively. Additionally, in all clones the inhibition of the cytopathic effect was 100% at a concentration of 40 μg/mL?
Response: If the result of the calculation (Percentage of without CPE higher than 50%−50%)/(Percentage of without CPE higher than 50%−Percentage of without CPE lower than 50%) is greater than 50%, it is considered effective.
13.LINES 261-262: Did the PBS (control) group also show amplification for BRV? So what difference is there between the two groups, if the control also tested positive?
Response: Based on the presented data, fecal BRV concentrations in the PBS group remained relatively stable over time. Although these values exceeded the lower limit of detection (approximately 3.4 log₁₀ copies/μL), this observed positivity in the PBS group may represent false signals which are attributable to the high sensitivity of RT-qPCR or potential laboratory contamination. In contrast, the BRV group exhibited slightly higher initial fecal BRV concentrations compared to the PBS group. However, when benchmarked against subsequent calf challenge data (not shown here), these values likely still fall within the negative range, as no significant temporal changes were observed. Throughout the observation period (12–156 h), BRV concentrations in both groups remained comparable, demonstrating stable and plateaued trajectories.
14.LINES 268-270: If damage was also observed in the control (PBS), how do you know that it was really BVR that caused the lesions in the organs? So what is the difference between both groups, if the control also tested positive? What do you associate that both groups showed damage in the intestine?
Response: In this experimental phase, the therapeutic effect was not evident in suckling mice. RT-qPCR results indicated comparable viral loads between the BRV and PBS groups, with no pathological changes observed, thus showing no difference in outcomes from the suckling mouse model. However, as both groups exhibited tissue damage, potential explanations include: (1) pre-existing intestinal conditions in the suckling mice (e.g., subclinical infections or stress responses) may have caused baseline damage even without viral intervention; or (2) PBS injection itself might have induced local tissue injury through mechanical stimulation (e.g., needle puncture) or aseptic inflammatory reactions, particularly in sensitive intestinal tissues.
15.LINES 289-290: were they not given any treatment for dehydration or pain management? Did all the animals in the group die? Were the animals left to die without any animal welfare? This is important to mention, since in your methodology you do not mention it. Although an ethics committee supposedly approved the protocol, there is NO SCIENTIFIC JUSTIFICATION for letting the animals die without humane slaughter.
Response: In this experimental study, due to the potential interference of analgesics with viral pathology and antibody efficacy, this consideration was explicitly addressed during ethical review. Following the onset of dehydration in calves, supportive interventions including twice-daily administration of saline solution and glucose were implemented to maintain electrolyte and energy homeostasis. However, due to sub-optimal therapeutic efficacy, the remaining subjects were humanely euthanized in accordance with pre-defined endpoints.
16.LINE 318 “in vitro” italics
Response: We sincerely thank the reviewer for their careful reading of the text. As suggested, we have corrected “in vitro” to “in vitro”.
17.LINES 354-361: The figure caption does not correspond with the images, when items A, B and C show different organ sections (duodenum, ileum, and jejunum) of the three experimental groups (BRV, PBS and Antibody)
Response: We sincerely thank the reviewer for their careful reading of the text. As suggested, we have corrected “Histopathological assessment” to “The results of tissue sectioning”.
18.LINE 368 “E.coli” italics
Response: We are sorry for our carelessness. In our resubmitted manuscript, this error has been addressed. Thank you for your correction.
19.LINES 404-405: Logarithmic numbers are expressed in superscript, please correct this.
Response: We are sorry for our carelessness. In our resubmitted manuscript, this error has been addressed. Thank you for your correction.
20.LINES 408: Scientific names are in italics.
Response: Thank you for your careful checking. We are sorry for our carelessness. Based on your comments, we have made the corrections to harmonize the use of italics throughout the whole manuscript.
21.LINES 408-418: not discussion. Please restructure the discussion
Response: We have re-written this part according to the Reviewer’s suggestion.
22.LINES 419-421 This is also mentioned in the methodology, mentioning it up to this point causes confusion.
Response: We have re-written this part according to the Reviewer’s suggestion.
23.LINES 433-438 was it inactivated, attenuated or live virus?
Response: We sincerely appreciate your valuable suggestions. The virus mentioned in the text regarding the issue you raised is an inactivated virus.
24.LINE 457 Correct the way of citing references
Response: We have re-written this part according to the Reviewer’s suggestion.
25.LINE 486 The scientific names are in italics.
Response: We have re-written this part according to the Reviewer’s suggestion.
26.LINE 488 Scientific names are in italics
Response: We have re-written this part according to the Reviewer’s suggestion.
27.Subsection 4.4: Where are the results of this section?
Response: Thank you for your suggestion. The results of section 4.4 correspond to subsection 2.4, as well as Figure 2C and Figure 2D.
28.Subsection 4.7: Lines 419-421 could be placed here.
Response: We sincerely appreciate your valuable suggestions. Regarding the issue you raised, we have made the necessary modifications to the text.
29.LINE 568: how was the PBS administered? It is important to mention how it was administered and the amount administered.
Response: PBS was orally administered to the PBS group at a dose of 100 μL/g, according to the weight of the rats at 0 h.
30.Subsection 4.8: the results of this section?
Response: Thank you for your suggestion. The results of section 4.8 correspond to Figure S3 in the supplementary materials.
31.LINE 578 Latin terms are in italics
Response: We have re-written this part according to the Reviewer’s suggestion.
32.Subsection 4.10. There is no mention of this test being approved under an ethics committee, is it the same as lines 445-450?
Response: Thank you for your suggestion. This question has already been answered in the article: Euthanasia was achieved by cervical dislocation for the suckling rats and by injection of saturated potassium chloride under deep anesthesia for the calves (approval number: NND2022023).
33.LINES 628-629 Were all animals in the experimental groups humanely killed or how many were killed? It is important to mention this and whether it was approved under an ethics committee.
Response: The calves that died during the experiment were dissected and sampled immediately after death. Except for the animals that died during the experiment, the animals in the PBS group, BRV group, and antibody group in other experiments were all euthanized at the end of the experimental period. This process was carried out in accordance with the guidelines and regulations of relevant institutions and the state, and was approved by the Experimental Animal Use and Care Committee of Inner Mongolia Agricultural University.
34.LINES 651-652 Is this antibody anti-bovine? It is also important to mention the time periods in which the stability of the recombinant antibody was monitored.
Response: Thank you for your suggestion. This antibody is specifically targeted against BRV, so it is an antibody for combating bovine diseases. After it reacts with bovine serum, we can observe the stability of the antibody within the bovine body. The time periods for detecting the stability of the recombinant antibody were 0, 3, 6, 12, 24, and 48 hours.
35.LINES 658-660 It is important to specify which variables or tests were statistically analyzed.
Response: As shown in the article, statistical analysis was applied to the results shown in graph 5D, as mentioned in subsection 2.8.2.
Round 2
Reviewer 1 Report
Comments and Suggestions for Authors
I thank the authors for their considerable efforts in revising the manuscript. The revised version shows substantial improvements in scientific focus, clarity, and organization, particularly in the introduction, discussion, and several key methodological descriptions. I appreciate the attention given to many of my original comments and acknowledge the significant progress made. However, upon careful review, several points still appear to be insufficiently addressed and should be revisited prior to final publication, except where clearly indicated as optional. I encourage the authors to conduct a final review round with these last few comments in mind.

Reviewer 3 Report
Comments and Suggestions for Authors
All my comments were considered in the improve version.
I have no further comments.
Author Response
Thank you very much for taking the time to review this manuscript. Please find the detailed responses below and the corresponding revisions / with track changes in the re-submitted files.
Thank you very much for your evaluation and reply. Thank you again for your review.